# High-Performance Dual-Redox-Mediator Supercapacitors Based on Buckypaper Electrodes and Hydrogel Polymer Electrolytes

**DOI:** 10.3390/polym16202903

**Published:** 2024-10-15

**Authors:** Garbas A. Santos Junior, Kélrie H. A. Mendes, Sarah G. G. de Oliveira, Gabriel J. P. Tonon, Neide P. G. Lopes, Thiago H. R. da Cunha, Mario Guimarães Junior, Rodrigo L. Lavall, Paulo F. R. Ortega

**Affiliations:** 1Grupo de Estudos em Dispositivos de Armazenamento de Energia (GEDAE), Departamento de Química, Universidade Federal de Viçosa, Viçosa 36570-900, Brazil; garbas.junior@ufv.br (G.A.S.J.); gabriel.tonon@ufv.br (G.J.P.T.); neide.lopes@ufv.br (N.P.G.L.); 2Centro Federal de Educação Tecnológica de Minas Gerais, Belo Horizonte 30421-169, Brazil; kelrie.mendes@gmail.com (K.H.A.M.); sarah_gabriela93@hotmail.com (S.G.G.d.O.); mgjunior@cefetmg.br (M.G.J.); 3Centro de Tecnologia em Nanomateriais e Grafeno—CTNano, Universidade Federal de Minas Gerais, Belo Horizonte 31310-260, Brazil; thiagocunha@ctnano.com.br (T.H.R.d.C.); rodrigo.lavall@qui.ufmg.br (R.L.L.); 4Departamento de Química/ICEx, Universidade Federal de Minas Gerais, Belo Horizonte 31270-901, Brazil

**Keywords:** buckypaper, supercapacitor, carbon nanotubes, hydrogel polymer electrolyte, redox mediator

## Abstract

In recent years, the demand for solid, thin, and flexible energy storage devices has surged in modern consumer electronics, which require autonomy and long duration. In this context, hybrid supercapacitors have become strategic, and significant efforts are being made to develop cells with higher energy densities while preserving the power density of conventional supercapacitors. Motivated by these requirements, we report the development of a new high-performance dual-redox-mediator supercapacitor. In this study, cells were constructed using fully moldable buckypapers (BPs), composed of carbon nanotubes and cellulose nanofibers, as electrodes. We evaluated the compatibility of BPs with hydrogel polymer electrolytes, based on 1 mol L^−1^ H_2_SO_4_ and polyvinyl alcohol (PVA), supplemented with different redox species: methylene blue, indigo carmine, and hydroquinone. Solid cells were constructed containing two active redox species to maximize the specific capacity of each electrode. Considering the main results, the dual-redox-mediator supercapacitor exhibits high energy density of 32.0 Wh kg^−1^ (at 0.8 kW kg^−1^) and is capable of delivering 25.9 Wh kg^−1^ at high power demand (4.0 kW kg^−1^). Stability studies conducted over 10,000 galvanostatic cycles revealed that the PVA polymer matrix benefits the system by inhibiting the crossover of redox species within the cell.

## 1. Introduction

Among traditional electrochemical energy storage systems, supercapacitors (SCs) are particularly strategic for applications requiring high power demands. These devices excel in meeting peak power requirements and offer a significantly longer lifespan compared to secondary batteries, particularly insertion batteries, and fuel cells [1]. SCs are prominently utilized in regenerative braking systems, power backup solutions, uninterruptible power systems (UPS), and the Internet of Things (IoT), where their unique advantages are most impactful [2].

Energy density is the primary limitation of SCs. The conventional cells developed in the 1990s and the early 2000s predominantly utilized porous electrodes, accumulating charge mainly through the electrical double layer. These cells exhibited energy densities not exceeding 10 Wh kg^−1^ [3]. During this period, transition metal oxides and conductive polymers also showed potential due to their pseudocapacitance, resulting from surface chemical reactions [4,5]. However, these materials suffer from chemical instability, which limits the cyclability of the system.

In recent years, alternative strategies have been explored to enhance the energy density of SCs. One approach involves combining ionic insertion electrodes (from battery technology) with double-layer capacitive electrodes, creating cells with hybrid charge accumulation [6]. These hybrid devices, however, are constrained in terms of power and cyclability by the battery-type materials [7]. Another promising strategy is the incorporation of redox mediators (redox additives) [8,9]. Hybrid SCs can be constructed by dissolving redox-active species in the electrolyte at high concentrations. The electrodes in these systems are typically composed of porous carbons [10], metal–organic frameworks [11], and other materials commonly used in conventional SCs [9]. In addition to storing electrostatic charge via an electrical double layer, the electrodes adsorb and promote the transfer of electrons from the redox species.

The success of these systems depends on several critical factors, including the adsorption capacity of the electrodes, compatibility with the redox mediators, solubility and chemical stability of the species [12], and their formal potential [9]. It is important to note that numerous redox species have been studied in various electrolytes, including aqueous, organic, and ionic liquids [9]. Moreover, SCs based on redox additives have already been explored in commercial-grade cells, demonstrating their practical applicability and potential for widespread use [13].

Recently, Alvarenga et al. demonstrated that paper-like films, also known as buckypapers (BPs), composed of carbon nanotubes (CNTs) and cellulose nanofibrils (CNFs), serve as exceptional electrodes when paired with electrolytes containing redox mediators [14]. The incorporation of CNFs enhances the hydrophilicity and tensile strength of the BPs. Santos et al. also demonstrated that these BPs can mediate reactions involving various species such as hydroquinone, iodide, and potassium hexacyanoferrate(II) [15]. These studies primarily focused on the preparation and physical properties of BPs. Additionally, the electrochemical performance of BPs has only been assessed in three-electrode cells using liquid electrolytes.

CNT/CNF-based BPs are completely moldable and open up possibilities for the construction of thin and flexible cells [16]. Electrochemical systems with these characteristics have been increasingly in demand for emerging applications ranging from modern displays to wearable devices. To achieve this, solid electrolytes must also be used to enable the assembly of a solid-state system [17,18]. It is in this context that the present work envisions exploring the application of these BPs in solid cells.

In this study, the electrochemical performance of BPs was investigated using polyvinyl alcohol (PVA)-based hydrogel polymer electrolytes (HGPEs) containing redox mediators. We evaluated different compositions of HGPEs prepared on BPs and examined the effect of adding organic electroactive species on the cells’ charge accumulation capacity. Finally, a dual-redox-mediator supercapacitor was constructed using methylene blue and indigo carmine within the HGPEs, each in contact with a different electrode.

It is important to highlight that few studies have explored cells with two redox mediators, primarily due to the complexity of the system, the potential for species crossover between electrodes, and the subsequent risk of self-discharge. Many of these studies are only feasible through the use of costly ion-selective membranes. This work, therefore, presents a proof of concept for a system with significant advantages, such as a dual-redox solid-state full cell assembled with thin, flexible electrodes, and without requiring an ion-selective membrane. Our data reveal that the PVA polymer matrix inhibits the crossover of redox species within the system, while also demonstrating excellent performance in terms of specific capacity, specific energy, and power density.

## 2. Materials and Methods

### 2.1. Preparation of Buckypapers (Electrodes)

BPs were prepared from a water dispersion of few-walled carbon nanotubes (FWCNT; CTNANO/UFMG) and cellulose nanofibrils (CNF) from Bambusa vulgaris, following a previously published methodology [14]. FWCNT and CNF were mixed in a mass ratio of 1:1, homogenized with a mechanical stirrer at 20,000 rpm for 25 min, and the resulting paste was spread on filter paper. After drying for 3 h at 120 °C, the paper-like composite was easily detached from the filter paper. Figure 1a,b shows a schematic and an image of the BP, respectively. All mechanical, and thermal, as well as textural characterizations and other physical–chemical properties of this BP are discussed in a previous work [14].

### 2.2. Preparation of Hydrogel Polymer Electrolytes and Cell Assembly

The hydrogel polymer electrolytes (HGPEs) were prepared by dissolving different masses (0.2, 0.4, 0.6, 0.8, and 1.0 g) of poly(vinyl alcohol) (PVA, Sigma-Aldrich, São Paulo, Brazil, Mw = 89,000–98,000) in 10 mL of 1 mol L^−1^ H_2_SO_4_ solution (Merck, São Paulo, Brazil, 98%). The solutions were then dripped (500 μL) onto the BPs cut into rectangles (2 cm × 1 cm) and subjected to the freeze–thaw method. The nomenclature used to differentiate the HGPEs resulting from this process was based on the masses used in the preparation of the solutions: 0.2-HGPE, 0.4-HGPE, 0.6-HGPE, 0.8-HGPE, and 1.0-HGPE. Five freeze–thaw cycles were carried out, with each freezing step at −5 °C for 2 h, followed by heating to 25 °C. Figure 1c shows a schematic of the HGPE buckypaper preparation process. After complete gelation of the electrolyte over the BP (Figure 1d shows a photography of the HGPE buckypaper), symmetrical cells were constructed by stacking two sheets of BP, with the HGPEs in contact, eliminating the need for conventional separator (the HGPEs also act as separators in this system). Figure 1e presents a schematic representation of the cell configuration, and Figure 1f shows a photograph of the flexible solid-state supercapacitor device. The symmetric supercapacitors based on HGPEs with different PVA amounts had some of their electrochemical properties compared to the same system configuration with an aqueous electrolyte (H_2_SO_4_, 1 mol L^−1^), referred to as the liquid electrolyte system in this paper.

To study solid cells containing redox additives, three different compounds were dissolved in the polymer solution (0.8-HGPE) before the gelation process. The compounds used were indigo carmine (IC; 0.05 mol L^−1^), methylene blue (MB; 0.05 mol L^−1^), and hydroquinone (HQ; 0.4 mol L^−1^). The concentrations were chosen to be close to the saturation of each compound in the polymer solution. The assembly of cells containing these dissolved molecules followed the same type of stacking method previously described (Figure 1g–i).

### 2.3. Electrochemical Tests

The electrochemical tests were conducted using an SP300 potentiostat (Biologic, Knoxville, TN, USA). To minimize measurement impedance, the cells were placed on gold plates within a PTFE Swagelok-type support. The cells underwent cyclic voltammetry measurements at various scan rates (10–100 mV s^−1^). Galvanostatic measurements at different current densities were employed to evaluate the efficiency (ε), specific capacity (*Q_sp_*), specific energy (*E_sp_*), and specific power (*P_sp_*) of the cells using Equations (1)–(4).
(1)$$\varepsilon =\frac{\Delta t_{discharge}}{\Delta t_{charge}}\times 100$$
(2)$$Q_{sp}=\frac{I\cdot \Delta t_{discharge}}{m\;}$$
(3)$$E_{sp}=\frac{I\int V\;dt}{m}\;$$
(4)$$P_{sp}=\frac{E_{sp}}{\Delta t_{\mathrm{discharge}}}\;$$
where *I* is the current applied, *t* is the time, *m* is the total mass of the electrodes, and *V* is the electrode potential (in 3-electrode measurements) or cell voltage (in 2-electrode measurements).

*Q_sp_* was the parameter used to evaluate the amount of charge stored in the electrode or cell, normalized by the mass of the electrodes (C g^−1^). ε represents the coulombic efficiency, which is the ratio of electrons extracted during the discharge stage to those consumed during the charge stage. This parameter is crucial for assessing the system’s stability. The energy (*E_sp_*) and power (*P_sp_*) densities were calculated for the complete cell, normalized gravimetrically by the electrode mass, with units of Wh kg^−1^ and W kg^−1^, respectively.

Electrochemical impedance spectroscopy measurements were conducted in a frequency range between 100 kHz and 50 mHz, with an amplitude of 10 mV, at open circuit potential. The experiments were conducted in 2- and 3-electrode cells. For 3-electrode measurements, a Ag pseudoreference was inserted between the HGPEs during the stacking of the BPs.

## 3. Results and Discussion

In a previous publication [14], we discussed the preparation and properties of the BP used as an electrode in this work. The inclusion of cellulose nanofibers renders BPs hydrophilic and mechanically robust, two desirable traits for solid and flexible device construction. Additionally, these BPs boast a thin profile (approximately 0.6 mm), complete moldability, and interparticle mesoporosity resulting from the entanglement between FWCNTs and CNFs. On the other hand, BPs face a significant limitation regarding their specific surface area (59 m^2^ g^−1^), which does not compare favorably to carbon materials such as activated carbons and graphenes that can easily exceed 1000 m^2^ g^−1^. Consequently, BP electrodes are restricted in their charge accumulation capabilities via the electric double-layer (capacitive) mechanism. However, these limitations can be mitigated through the incorporation of redox additives, which significantly enhance the charge accumulation capacity of the system through faradaic reactions. BPs exhibit a high adsorption capacity for redox mediators, as demonstrated by Santos et al. using compounds such as K_4_[Fe(CN)_6_], hydroquinone, and KI [15].

In this study, we initially constructed symmetric cells using HGPEs with varying concentrations of PVA gelled onto the surface of BPs, without any dissolved redox species. A higher concentration of PVA promotes electrolyte gelation with enhanced mechanical stability, minimizing the risk of leakage and thus improving device integrity [17]. However, in polymer electrolytes, the ionic mobility of the species is reduced because the ions are confined within the polymer network and interact with the polymer chains. Additionally, the concentration of ionic species decreases at the interface, affecting the structure of the electric double layer. Consequently, the charge accumulation capacity of cells utilizing HGPE is lower than that of cells employing a liquid electrolyte. This effect is even more pronounced in electrolytes with higher polymer fractions, which can further increase contact resistance at the electrode/electrolyte interface.

To determine the optimal composition among the investigated HGPEs, we conducted electrochemical impedance spectroscopy (EIS) and galvanostatic charge–discharge (GCD) measurements in two-electrode cells (BP|HGPE|BP). Figure 2a presents the Nyquist diagrams with an emphasis on the high–medium frequency region. A circuit of the form R_A_(Q[R_ct_W]) was used to model the experimental data, where R, Q, and W correspond to the elements of resistance, constant phase element, and Warburg impedance, respectively.

R_A_ is associated with the sum of the resistances of the components (electrode + electrolyte + current collectors) and contacts. When the cell is built with a liquid electrolyte (using conventional glass fiber separators), this resistance is the lowest (R_A_ = 0.38 Ω). Using HGPEs, the R_A_ values gradually increase with the mass fraction of the polymer, reaching up to 2.31 Ω for 1.0-HGPE. Another important element is R_ct_, which corresponds to the charge transfer resistance and is associated with the pseudocapacitive reactions occurring on the surface of the electrodes. The origin of these chemical reactions arises from the oxygenated groups present on the FWCNT and CNFs [15,19]. It is important to note that all charge transfer must be accompanied by the mass transfer of ions in the electrolyte, which is affected by the higher polymer mass fraction. As the polymer fraction decreases, the concentration of ions at the interface and the compensation for charge variation associated with redox processes also decrease. Consequently, the R_ct_ values of cells built with HGPEs increase in the following order: 0.2-HGPE (0.25 Ω) < 0.4-HGPE (1.04 Ω) < 0.6-HGPE (1.29 Ω) < 0.8-HGPE (2.44 Ω) < 1.0-HGPE (4.51 Ω). It should be noted that the greatest relative increase in R_A_ and R_ct_ values occurs between cells containing 0.8-HGPE and 1.0-HGPE, indicating that the polymer mass fraction in 1.0-HGPE has become excessive.

Galvanostatic measurements were used to evaluate the charge accumulation capacity of cells built with different HGPEs. Figure 2b presents the GCD curves obtained at a current density of 1 A g^−1^ (3.54 mA cm^−2^). In all cases, the charge accumulation mechanism is mostly capacitive, with small deviations from the triangular behavior typical of supercapacitors due to pseudocapacitive reactions (arising from oxygenated functional groups). The conventional cell containing the liquid electrolyte was again used as a reference and presented the highest specific capacity (Q_sp_) values (31.9 C g^−1^).

It is notable how the use of polymer electrolyte reduces the charge storage capacity by almost half. Polymer chains reduce the maximum amount of ions in the electric double-layer, in addition to reducing ionic mobility [20]. Despite this, using electrolytes with a composition between 0.4-HGPE and 0.8-HGPE, the Q_sp_ values remain relatively close, ranging between 14.5 and 14.1 C g^−1^. The largest reduction in capacity is observed for 1.0-HGPE (4.6 C g^−1^), corroborating the more resistive behavior observed in the EIS measurements.

At this point, it is important to highlight that BPs have a low specific surface area and, therefore, deliver low capacity compared to activated carbon, for example [21]. On the other hand, its excellent compatibility with redox mediators, which act as a capacity booster, has already been demonstrated using conventional liquid electrolytes [15]. To study different redox molecules (MB, HQ, and IC) in cells containing polymer electrolytes, only one HGPE composition was evaluated. In this case, 0.8-HGPE was chosen because it presents the highest maximum polymer fraction without substantially sacrificing the cell’s electrochemical performance. So, this is the electrolyte with the optimized composition. Initially, voltammetric measurements, in a three-electrode configuration, were carried out on 0.8-HGPE, Figure 3a, and then the study of the faradaic reactions of the MB, HQ, and IC on BPs, as shown in Figure 3b–d.

The three redox species dissolved in 0.8-HGPE can transfer two electrons in an acidic medium, as illustrated in Figure 3b–d. These processes can be identified in all CV curves, but the formal potentials are more separated in the cell containing IC, as shown in Figure 3d. Analyzing the most intense signals, the systems present a quasi-reversible behavior with the peak potential varying with the scan rate (υ). Therefore, the peak currents (i_p_) are controlled by both the mass transfer and charge transfer steps. Furthermore, the high correlation coefficient for i_p_ vs. υ^1/2^ (see the insert graph) demonstrates that the processes are controlled by diffusion for the three redox species. Considering the magnitude of the voltammetric currents, the increase in the faradaic current in relation to the capacitive one of the electrodes in contact with the HGPE without redox additive is notable. These results demonstrate that the 0.8-HGPE matrix does not limit the use of redox species as a capacity booster for the system.

The operating potentials of each redox species are important for the selection of the poles in a full cell. The MB has a half-wave potential for the maximum intensity signal centered at +0.220 V (vs. Ag/Ag^+^). For HQ, this potential is centered at +0.088, and for IC at −0.014 V. To construct the dual-redox-mediator SC, the pair of species with the greatest formal potential difference were selected, using MB and IC in the HGPEs in contact with the positive and negative electrodes, respectively. Figure 4a presents schematic representation of the arrangement of the components the solid cell.

Figure 4b presents the GCD curves at different current densities for the dual-redox-mediator SC. The inset graph shows a comparison with the GCD curves of the SC constructed without redox additive. With the addition of redox species, the system exhibits hybrid behavior, characterized by the appearance of plateaus at intermediate voltages due to faradaic processes. These processes provide greater capacity compared to capacitive ones, resulting in longer charge/discharge times for the cell. The voltammetric curves (obtained in the two-electrode configuration) also exhibit redox peaks with intensities two orders of magnitude higher than the capacitive current (Figure 4c). As a result of the incorporation of the two redox species, the specific capacity of the cells increased significantly. Figure 4d shows the specific capacity for the dual-redox-mediator SC compared to the 0.8-HGPE SC at different current densities. At the lowest current density evaluated (1 A g^−1^/3.54 mA cm^−2^), the dual-redox SC and conventional SC deliver 143.9 C g^−1^ (40.0 mAh g^−1^) and 14.1 C g^−1^ (3.9 mAh g^−1^), respectively. Considering the rate capability, with the increase in the applied current to 5 A g^−1^ (17.69 mA cm^−2^), the dual-redox SC reduces its charge accumulation capacity by approximately 81% (116.8 C g^−1^/32.4 mAh g^−1^).

Although the voltammetric and galvanostatic curves of the cells exhibit faradaic processes, these signals are complex and result from the contribution of both electrodes, with the reaction of the two redox species occurring simultaneously. Therefore, it is not possible to evaluate the extent of the reaction of each molecule and differentiate the performance of each electrode in a study of cells in the two-electrode configuration. To address this, galvanostatic measurements were also conducted with the insertion of a Ag pseudoreference in the gel electrolyte, allowing for the evolution of the potentials of each electrode to be recorded. Figure 5a presents the GCD curves for the positive and negative electrodes at different current densities.

Examining the behavior of each individual electrode, the positive electrode shows a small variation in ∆E throughout the cell charging/discharging due to the greater extent of the MB reaction. The average potential occurs at +0.220 V (vs. Ag/Ag^+^), in agreement with voltammetric measurements carried out in the three-electrode experiments, Figure 3b. Therefore, depletion of the MB concentration at the interface is not observed. For the negative electrode, the evolution of its potentials results in different zones, one with faradaic predominance and others dominated by the electric double-layer mechanism. This electrode is the limiting one, with the transfer of electrons from the IC being exhausted in the range between −0.02 and −0.2 V (vs. Ag/Ag^+^). As a result, its contribution to the capacity is much smaller in relation to the positive electrode. This finding is also interesting from a fundamental point of view, revealing that although the electrolytes are prepared with the same concentrations of MB and IC, the electrosorption of the MB cation species is greater on the BP interface compared to IC (anionic). Furthermore, this issue opens doors for future research, either by exploring combinations of other species or through the synthesis of new redox molecules that enhance electrosorption and exhibit low activation energy for electron transfer.

The cyclic stability of the dual-redox-mediator SC was also investigated through galvanostatic measurements at 2.5 A g^−1^ (8.84 mA cm^−2^), Figure 5b. Throughout the entire experiment, the cell’s coulombic efficiency remains in the range of 87 to 93%. In relation to the initial capacity (123 C g^−1^), the cell suffers a continuous decrease in performance (average reduction of 5.7 mC per cycle), reaching 66 C g^−1^ at the end of 10,000 cycles. Many factors can influence this behavior, including the gradual decomposition of the redox mediators and structural changes in the entanglement between CNFs and CNTs, which reduce the surface area and the number of sites available for electrosorption.

When analyzing the profile of the galvanostatic curves of the electrodes in the last cycle, another important benefit of using HGPE in the construction of the dual-redox SC becomes evident. It is noted that the operating potentials of the IC and MB redox species remained unchanged (Figure 5c). This indicates that the polymer matrix limits the diffusion and crossover of redox additives between the electrode/electrolyte interfaces. This significant effect was also observed by Wang et al. in the study of supercapacitors containing CuCl_2_-H_2_SO_4_-PVA gel electrolytes [22]. In that work, copper ions were used as redox additives and self-discharge was suppressed by the PVA polymer matrix. The importance of this result is further justified by the cost of the cells. Different studies exploring dual-redox-mediator-enhanced SCs employ cation or anion exchange membranes to prevent mixing of redox species [23,24,25].

The specific energy (E_sp_) and specific power (P_sp_) values achieved by the dual-redox SC are presented in the Ragone diagram depicted in Figure 5d. This diagram presents the optimal E_sp_ and P_sp_ values for various SC cells reported in the literature. It is noteworthy that the solid cell in this study exhibits excellent E_sp_ values (25.9 Wh kg^−1^) under high power demands (4.0 kW kg^−1^) while utilizing an HGPE. This outstanding performance is attributed to the structural advantages of BPs, which facilitate rapid charge transfer on their surface. Moreover, the achieved maximum specific energy (32.0 Wh kg^−1^ at 0.8 kW kg^−1^) places this system on par with other reported cells.

**Figure 5 polymers-16-02903-f005:**
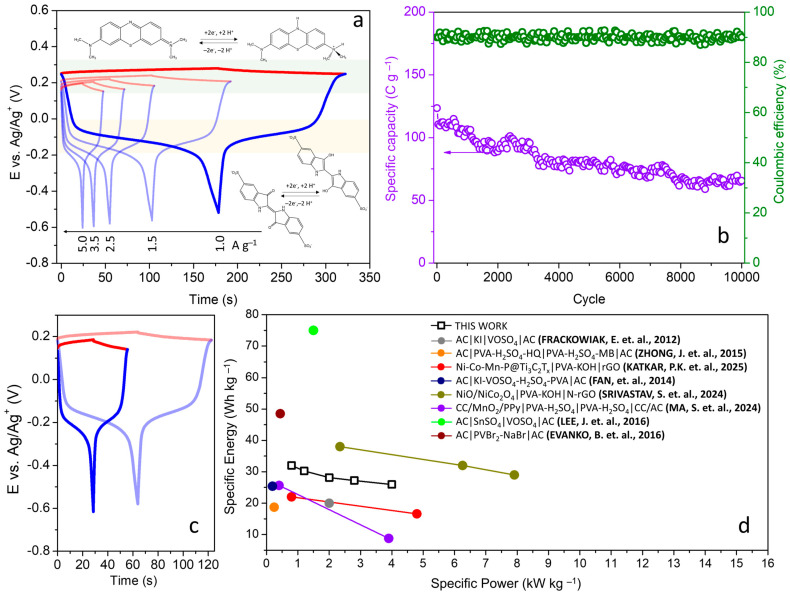
Dual-redox-mediator solid-state SC: (**a**) evolution of the potential of the electrodes at different current densities (red line, positive electrode (methylene blue redox mediator); blue line, negative electrode (indigo carmine redox mediator)); (**b**) cycling stability for 10,000 cycles at 2.5 A g^−1^ (8.84 mA cm^−2^) and coulombic efficiency; (**c**) comparison of the potential evolution of the electrodes at the 1st and the 10,000th cycle. (**d**) Ragone plot of the dual-redox-mediator solid-state SC, compared with some previously published systems [19,25,26,27,28,29,30,31].

Katkar, P. V. et al. developed a composite Ni-Co-Mn-P@T_3_C_2_T_x_ used as the cathode in a solid-state supercapacitor (SC). They used reduced graphene oxide (rGO) as the anode and PVA-KOH as the electrolyte. The SC, Ni-Co-Mn-P@ T_3_C_2_T_x_|PVA-KOH|rGO, delivered an energy density of 22.0 Wh kg^−1^ at a power density of 0.8 kW kg^−1^. At the same power density, our system delivered a higher energy density of 32.0 Wh kg^−1^ [26].

The double-redox-mediator SC developed here can also be compared to other systems employing electrodes with high specific surface areas, such as activated carbons (ACs). For example, Fan et al. presented a system, AC|KI-VOSO_4_-H_2_SO_4_-PVA|AC, containing double-redox actives, which delivered an energy density of 25.4 Wh kg^−1^ at a power density of 0.19 kW kg^−1^ [27]. On the other hand, systems with high surface area (235 m^2^ g^−1^) metal oxides, such as NiO/NiCo_2_O_4_|PVA-KOH|N-rGO, exhibit even higher energy density, achieving 38.0 Wh kg^−1^ at 2.35 kW kg^−1^, as expected [28].

Our system can also be compared to other works utilizing gel polymer electrolytes that employ membranes as separators. In this context, our work demonstrates higher electrochemical performance without the need for expensive membranes. For instance, Ma et al. investigated a polypyrrole (PPy) coated MnO_2_ nanosheet on carbon cloth (CC), denoted as CC/MnO_2_/PPy, used as the cathode, and CC/AC as the anode. Using a PVA-H_2_SO_4_ gel, the supercapacitor was assembled with a non-woven diaphragm serving as the separator. The configuration CC/MnO_2_/PPy|PVA-H_2_SO_4_|PVA-H_2_SO_4_|CC/AC delivered an energy density of 25.67 Wh kg^−1^ at a power density of 0.88 kW kg^−1^ [29]. Additionally, a Nafion 117 membrane was employed as a separator in a dual-redox mediator supercapacitor, AC|PVA-H_2_SO_4_-HQ|PVA-H_2_SO_4_-MB|AC, which delivered an energy density of 18.7 Wh kg^−1^ at a power density of 0.24 kW kg^−1^ [25].

Finally, this system stands out among other dual-redox supercapacitors (SCs) utilizing liquid electrolytes. For example, the configuration AC|KI|VOSO_4_|AC (with a Nafion membrane) delivered an energy density of 20.0 Wh kg^−1^ at a power density of 2.0 kW kg^−1^. However, some other aqueous systems exhibit higher electrochemical performance. For instance, AC|SnSO_4_|VOSO_4_|AC (with a Fas15 anion exchange membrane) delivered an energy density of 75.0 Wh kg^−1^ at a power density of 1.5 kW kg^−1^ [30], and AC|PVBr_2_-NaBr|AC achieved an energy density of 48.5 Wh kg^−1^ at a power density of 0.44 kW kg^−1^ [31]. These results demonstrate that the dual-redox-mediator SC constructed in this work is competitive with a variety of different systems reported, considering its high energy density and power.

## 4. Conclusions

In this work, we present a new dual-redox-mediator supercapacitor (SC) utilizing BP electrodes. To optimize the electrochemical performance of solid-state cells, we investigated the impact of polyvinyl alcohol (PVA) mass fraction on the impedance and capacity of BP electrodes. At the optimal composition of 0.8-HGPE, the hydrogels were enhanced with organic redox mediators. Our results showed that the polymer matrix does not hinder electron transfer from the redox-active species to the BPs. Consequently, in solid-state cells constructed with two redox mediators, the specific capacity was ten times higher (143.9 C g^−1^ at 1 A g^−1^) compared to conventional SCs (14.1 C g^−1^ at 1 A g^−1^). Furthermore, the maximum energy and power density values achieved by the dual-redox-mediator SC (32.0 Wh kg^−1^ and 4.0 kW kg^−1^, respectively) are among the highest reported for hybrid systems based on aqueous electrolytes. Additionally, our findings highlight another significant advantage of using PVA hydrogels as electrolytes: they mitigate the crossover of redox species in the cell over 10,000 galvanostatic cycles, thus enabling the construction of durable solid-state cells.

As a proof of concept, this work offers a glimpse into future perspectives, particularly in optimizing the limiting electrodes of the full cell and improving capacity retention throughout cycling, which was 53% for this dual-redox-mediator SC. New combinations of mediators could be explored to achieve electrodes with similar charge accumulation capacities. Moreover, the synthesis of new redox species with high solubility and enhanced chemical stability should be considered not only for capacity and energy density improvements, but also to ensure cyclic stability approaching that of electrochemical double-layer capacitors. Furthermore, the evaluation of different polymer matrices’ ability to inhibit redox species crossover has not yet been systematically investigated, which is crucial for the final device’s cost.

## Figures and Tables

**Figure 1 polymers-16-02903-f001:**
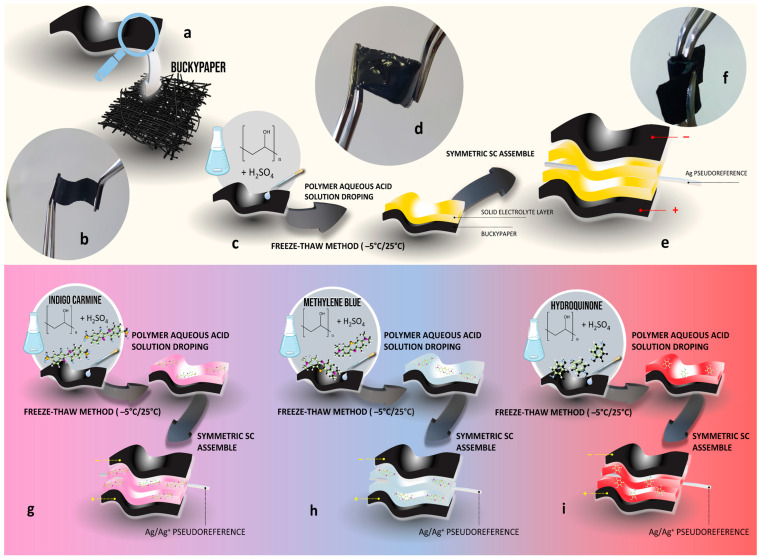
(**a**) Schematic representation and (**b**) image of the obtained CNT/CNF buckypaper. (**c**) Schematic of the HGPE buckypaper preparation process and (**d**) photography of the HGPE buckypaper. (**e**) Schematic representation of the cell configuration and (**f**) photography of the flexible solid-state supercapacitor device. Schematic representation of the fabricated symmetric SC device assembled using 0.8-HGPE buckypaper and redox mediator: (**g**) carmine indigo, (**h**) methylene blue, (**i**) hydroquinone.

**Figure 2 polymers-16-02903-f002:**
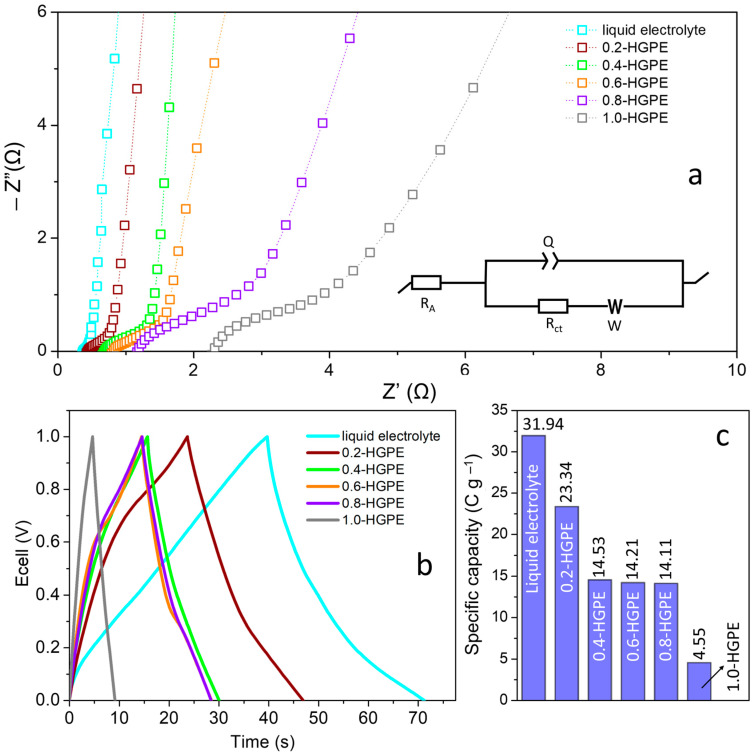
(**a**) Nyquist plot—inset shows the equivalent circuit; (**b**) GCD curves at J = 1 A g^−1^; (**c**) specific capacity of SCs based on HGPE electrolyte compared to the liquid electrolyte system at 1 A g^−1^ (3.54 mA cm^−2^).

**Figure 3 polymers-16-02903-f003:**
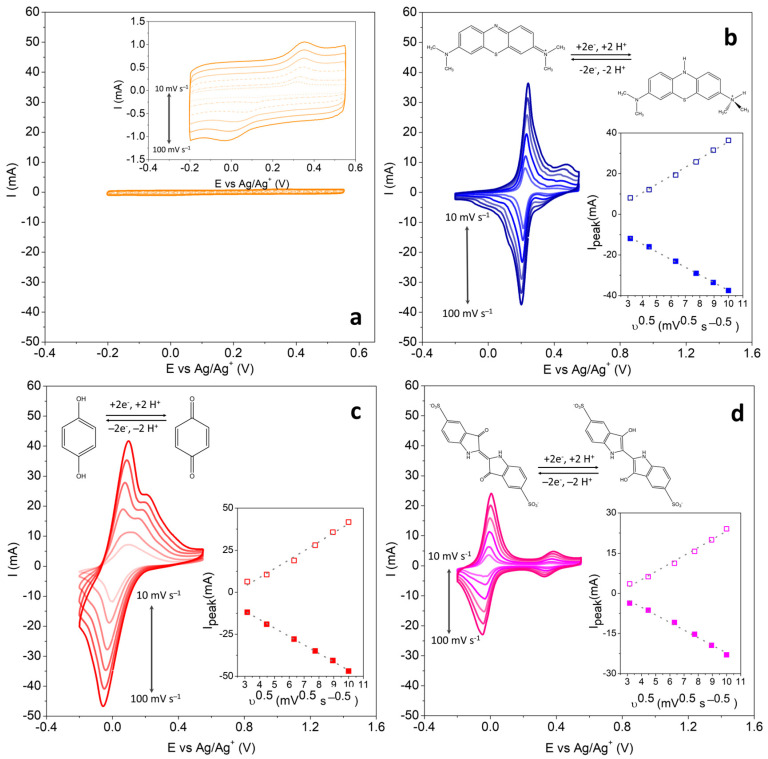
Cyclic voltammetry for cells constructed with (**a**) 0.8-HGPE—the inset shows the magnified cyclic voltammetry—and 0.8-HGPE containing (**b**) methylene blue, (**c**) hydroquinone, and (**d**) indigo carmine. The inset also shows the dependence of peak currents on the square root of the scan rate for both anodic and cathodic potentials. The anodic peak current is represented by empty squares, while the cathodic peak current is represented by filled squares.

**Figure 4 polymers-16-02903-f004:**
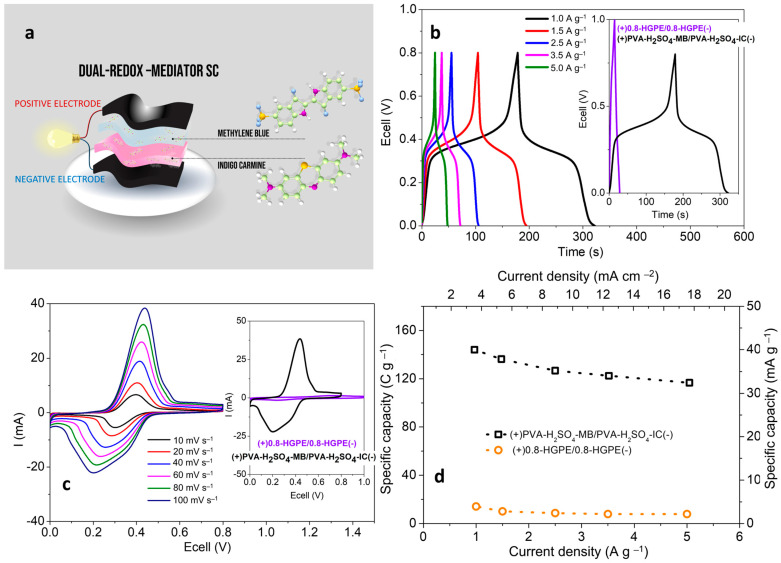
Dual-redox-mediator solid-state SC: (**a**) schematic configuration; (**b**) GCD curves at different current densities, inset compares the system with 0.8-HGPE SC, at 1 A g^−1^; (**c**) cyclic voltammetry at different scan rates, inset compares the system with 0.8-HGPE SC, at 100 mV s^−1^; (**d**) specific capacity compared to 0.8-HGPE SC.

## Data Availability

The data adopted in support of the findings in this work are available upon request from the corresponding author.

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
