# Peer review of "High-Performance Dual-Redox-Mediator Supercapacitors Based on Buckypaper Electrodes and Hydrogel Polymer Electrolytes"

_polymers, 2024, doi:10.3390/polym16202903_

Round 1

Reviewer 1 Report

Comments and Suggestions for Authors

The following comments are suggested to improve the quality of this manuscript further.

1. It could be overstated (in the abstract) to say the device of the studied shows close energy density comparing to the battery, because 20-30 Wh/kg is too low.

2. It is suggested to show the cycling performance of each redox mediator: redox species: methylene blue, indigo carmine, and hydroquinone, in the half cells.

3. In Figure 2a, it doesn't make sense to assign a Rct to a half-ellipse, which should be a semicircle.

4. Figure 5b, y-axis should start from a positive value, although that would be easy to observe the fast decay of the cycling stability.

5. The Figure 5c is confusing to read. Please replot the figure with clear labels/colours.

Reviewer 2 Report

Comments and Suggestions for Authors

Reviewer: Major Revision

After going through the manuscript, I think that the manuscript requires major improvements in the following areas:

1-    Carefully proofread the manuscript to correct typographical errors and formatting issues, such as inconsistent spacing and misaligned figure captions. Additionally, make sure all figures and tables are appropriately referenced in the text. Editing of English language is required.

2-    The introduction should be refined to address the novelty of the present work to the related reported ones. Further, some important definitions such as pseudocapacitance, quantum capacitance, total interfacial capacitance, differential EDL capacitance are missing. Authors must provide brief definitions of these important definitions when they first appear in the manuscript. The authors are recommended also to outline the unique contributions of this manuscript in bullet points within the introduction section. Additionally, they should include a brief overview of the manuscript's organization at the end of the introduction.

3-    After the introduction part, authors must insert new section for related works under title of "Literature review", detailing previous research in this field and its limitations. Authors can use the results of this section to compare with the results of the current research (One more Table is required). Authors must use suitable recent references.

4-    In the results part, there is no any characterization for the electrode material. Authors should add some characterization to proof the quality of the electrode material.

5-    More discussion about the Rct and the used circuit is reuired.

6-    The authors should add a section discussing the limitations of the current study. The conclusion should succinctly summarize the key findings and contributions. Additionally, the authors should highlight potential directions for future work.

7-    Generally, authors must pay more effort to explain them findings throughout the manuscript.

Comments on the Quality of English Language

Minor editing of English language is required.

Reviewer 3 Report

Comments and Suggestions for Authors

·        The report emphasises the electrode's low specific surface area (59 m2/g) as a limiting issue. This results in a lower capacity than other high surface area materials like activated carbon, which can have a major impact on overall energy storage efficiency.

·        Research indicates that even with redox-coupled mediators, negative electrode electron transfer (indigocarmine) contributes significantly less than positive electrode (methylene blue). This asymmetry in electron transmission restricts the system's capacity and may have an impact on the cell's overall performance.

·        The use of PVA hydrogel polymer electrolyte lowers ionic mobility and the amount of ions in the electrodouble layer. This significantly reduces the specific capacity. In comparison to traditional liquid electrolytes, this limits optimal charge storage.

·        This article shows that cycling causes a constant decline in specific capacity. After 10,000 cycles, capacity drops by around 50% (from 123 to 66 C/g). This decrease in capacity may impair stability and reliability.

·        Studies indicate that the highest polymer mass fraction (1.0-HGPE) results in high ionic resistance and diminish charge transfer efficiency. This results in a severe decrease of the system's distinctive capabilities. This suggests that the electrolyte structure has to be further optimised.

The above comments are important to improve the quality of the article. Please provide thoughtful comments and edit the report accordingly.

Round 2

Reviewer 2 Report

Comments and Suggestions for Authors

Reviewer: Accepted

Authors have been revised and corrected reviewer's comments. I suggestion this manuscript can be accepted for publication in your journal in this status.

Reviewer 3 Report

Comments and Suggestions for Authors

The authors have addressed all the queries, and the paper can be accepted.